# SARS CoV-2 (Delta Variant) Infection Kinetics and Immunopathogenesis in Domestic Cats

**DOI:** 10.3390/v14061207

**Published:** 2022-06-01

**Authors:** Miruthula Tamil Selvan, Sachithra Gunasekara, Ping Xiao, Kristen Griffin, Shannon R. Cowan, Sai Narayanan, Akhilesh Ramachandran, Darren E. Hagen, Jerry W. Ritchey, Jennifer M. Rudd, Craig A. Miller

**Affiliations:** 1Department of Veterinary Pathobiology, College of Veterinary Medicine, Oklahoma State University, Stillwater, OK 74078, USA; miruthula.tamil_selvan@okstate.edu (M.T.S.); sachithra.gunasekara@okstate.edu (S.G.); kristen.griffin@okstate.edu (K.G.); shannon.r.cowan@okstate.edu (S.R.C.); jerry.ritchey@okstate.edu (J.W.R.); jennifer.rudd@okstate.edu (J.M.R.); 2Department of Animal and Food Sciences, Ferguson College of Agriculture, Oklahoma State University, Stillwater, OK 74075, USA; ping.xiao@okstate.edu (P.X.); darren.hagen@okstate.edu (D.E.H.); 3Oklahoma Animal Disease Diagnostic Laboratory, College of Veterinary Medicine, Oklahoma State University, Stillwater, OK 74078, USA; ssankar@okstate.edu (S.N.); rakhile@okstate.edu (A.R.)

**Keywords:** SARS-CoV-2, Delta variant, B.1.617.2, feline, domestic cats, pathology, inflammatory pathways, disease transmission

## Abstract

Continued emergence of SARS-CoV-2 variants highlights the critical need for adaptable and translational animal models for acute COVID-19. Limitations to current animal models for SARS CoV-2 (e.g., transgenic mice, non-human primates, ferrets) include subclinical to mild lower respiratory disease, divergence from clinical COVID-19 disease course, and/or the need for host genetic modifications to permit infection. We therefore established a feline model to study COVID-19 disease progression and utilized this model to evaluate infection kinetics and immunopathology of the rapidly circulating Delta variant (B.1.617.2) of SARS-CoV-2. In this study, specific-pathogen-free domestic cats (*n* = 24) were inoculated intranasally and/or intratracheally with SARS CoV-2 (B.1.617.2). Infected cats developed severe clinical respiratory disease and pulmonary lesions at 4- and 12-days post-infection (dpi), even at 1/10 the dose of previously studied wild-type SARS-CoV-2. Infectious virus was isolated from nasal secretions of delta-variant infected cats in high amounts at multiple timepoints, and viral antigen was co-localized in ACE2-expressing cells of the lungs (pneumocytes, vascular endothelium, peribronchial glandular epithelium) and strongly associated with severe pulmonary inflammation and vasculitis that were more pronounced than in wild-type SARS-CoV-2 infection. RNA sequencing of infected feline lung tissues identified upregulation of multiple gene pathways associated with cytokine receptor interactions, chemokine signaling, and viral protein–cytokine interactions during acute infection with SARS-CoV-2. Weighted correlation network analysis (WGCNA) of differentially expressed genes identified several distinct clusters of dysregulated hub genes that are significantly correlated with both clinical signs and lesions during acute infection. Collectively, the results of these studies help to delineate the role of domestic cats in disease transmission and response to variant emergence, establish a flexible translational model to develop strategies to prevent the spread of SARS-CoV-2, and identify potential targets for downstream therapeutic development.

## 1. Introduction

The swift emergence of the B.1.617.2 (Delta) variant of severe acute respiratory syndrome coronavirus-2 (SARS-CoV-2) prompted urgent investigation to better understand why this variant so efficiently outcompeted previous variants of concern. Questions as to Delta variant’s transmissibility, replication efficiency, clinical severity, and ability to evade immunity had to be answered quickly in order to mitigate resulting damage. Studies showed that Delta variant gained a notable replication advantage and had significantly reduced sensitivity to neutralizing antibodies compared to B.1.1.7 (Alpha) [1]. In addition, viral loads of Delta variant infections were about 1000 times greater than previous lineages [2] and Delta variant infection was associated with an increased risk of hospitalization, especially among the unvaccinated [3,4]. Despite these advances in knowledge, Delta variant’s advantages in virulence, shortened incubation periods, high viral loads and longer duration in shedding as compared with the original wild-type SARS-CoV-2 [2,5,6,7,8] have resulted in at least 4.1 million deaths globally. Even as cases of Delta variant decline, a deeper understanding of the complex infection kinetics, impact of immune dysfunction, and transmission potential of the SARS-CoV-2 Delta variant has major implications in our potential response to new emerging variants of concern (VOCs).

Delta variant’s rapid rise to dominance underscored an urgency to have validated animal models that will continue to adapt over time to mimic the course of human SARS-CoV-2 infections and disease. Translational animal models are a critical step in delineating immunopathogenesis of disease and developing and testing mitigation strategies and vaccine prototypes prior to human cohort studies, and this is especially crucial under recent circumstances where prior vaccine- or infection-induced immunity does not fully impart cross-protective immunity to newly arising VOCs [9,10,11]. Animal models commonly used for SARS-CoV-2 studies include mice, ferrets, hamsters, mink, non-human primates, cats, and deer [12,13,14,15,16]. However, the utility of these animal models is often limited by one or more factors such as a lack of natural viral-entry receptors (wild-type mice), lack of severe pulmonary disease and acute respiratory distress syndrome (ARDS) (huACE2 transgenic mice, hamsters, ferrets), scarce availability and zoonotic risks (non-human primates), or difficulty in acquiring and handling specific pathogen free (SPF) animals (mink). Domestic cats, on the other hand, offer a tractable model to study COVID-19 as they are (i) naturally infected with SARS-CoV-2 through ACE2 binding, (ii) exhibit clinical signs of respiratory illness and systemic disease during acute infection, and (iii) can transmit the virus from cat to cat [17,18,19,20,21,22,23,24,25,26].

The natural susceptibility of cats to SARS-CoV-2 is a result of widespread and similar distribution of feline ACE2 receptors as in humans [27], and our previous studies of wild-type SARS-CoV-2 infection in domestic cats have provided insights into the impacts of feline ACE2 receptor distribution on pathologic lesion development [26]. While early studies documented only mild pulmonary disease and no clinical signs of illness with intranasal and/or intraoral inoculation of wild-type SARS-CoV-2 in domestic cats [23,27,28,29], our more recent studies found that intratracheal inoculation of wild-type SARS-CoV-2 produced significant clinical disease (lethargy, fever, dyspnea, and dry cough) in cats and pulmonary lesions (diffuse alveolar damage, hyaline membrane formation) consistent with the early exudative phase of COVID-19 in humans [26]. Furthermore, transmission of SARS-CoV-2 between domestic cats and to domestic cats and other felids from owners or caretakers is well-documented [30,31,32,33], and while zoonosis from felids to humans is not yet documented, the potential persists, especially with continued emergence of new variants. Moreover, infection kinetics and transmission potential of major VOCs (such as the SARS-CoV-2 Delta variant) have not been well studied in domestic cats. Investigating key factors contributing to COVID-19 progression in cats can therefore have a substantial impact on understanding mechanisms of immune dysfunction and disease progression in humans—especially since comorbidities (hypertension, diabetes, renal disease and obesity) that exacerbate COVID-19 disease are common in cats and readily adapted to feline models.

The similarities in infection kinetics and pathology between acute COVID-19 in cats and people, coupled with significant gaps in our understanding of how newer variants affect infection kinetics and disease, highlight the potential for domestic cats to serve as an important translational model for understanding pathogenesis of SARS-CoV-2 in both animals and humans. This study aims to not only determine the susceptibility of domestic cats to Delta variant of SARS-CoV-2, but also to assess the adaptability of this domestic cat model to new variations of COVID-19 in people. Using this model to better understand immune dysfunction, infection kinetics, and clinical disease and pathology offers insight into the immunopathogenesis of infection in hospitalized patients with COVID-19. Elucidating these key aspects of disease in the feline model also allows cats to be a translational pathway to test new pharmacologic agents and treatment combinations that can directly affect clinical outcomes in human patients.

## 2. Materials and Methods

### 2.1. Cells and Virus

Vero (CCL-81) cells (ATCC, Manassas, VA, USA) were used for viral propagation and titration. The SARS-CoV-2 viral strain B.1.617.2-hCoV-19/USA/PHC658/2021 (Delta variant) was obtained from BEI Resources (Manassas, VA, USA) and passaged up to seven times in Vero cells in Dulbecco’s Modified Eagle Medium (Gibco, Carlsbad, CA, USA) with 5% fetal bovine serum (Hyclone, Logan, UT, USA) and antibiotics. Viral stock was collected and TCID50 was calculated in Vero cells using the Reed and Muench method [34] as previously described [26].

### 2.2. Animals

A total of thirty (*n* = 30) age-matched (1-year-old), sex-matched (15 male, 15 female), spayed/neutered specific pathogen-free (SPF) cats were purchased from Marshall Bioresources (North Rose, NY, USA). Animals purposed for SARS CoV-2 inoculation (*n* = 24 total) were contained within Animal Biosafety Level 3 (ABSL-3) barrier animal rooms at Oklahoma State University (Stillwater, OK, USA) for the duration of the study and fed dry/wet food with access to water ad libitum [26]. Animals intended for sham-inoculation (negative controls, *n* = 6) were group-housed within an AAALAC International-accredited animal facility at OSU. All cats were examined clinically for their health status and were allowed to acclimate for 30 days prior to the initiation of the study. Temperature sensing microchips (Bio Medic Data Systems, Seaford, DE, USA) were subcutaneously implanted in the dorsum as previously described [26]. Body weights, temperatures, and nasal swab samples were obtained prior to inoculation, and all cats were considered in good health at the onset of the study.

### 2.3. Virus Challenge

The study was conducted in two phases based upon the duration of SARS-CoV-2 infection, and cats were divided equally for each phase as follows. Phase 1 consisted of a 4-day study utilizing a total of 15 SPF cats: *n* = 12 SARS-CoV-2-infected and *n* = 3 uninfected (negative control). Phase 2 consisted of a 12-day study also utilizing a total of 15 SPF cats: *n* = 12 SARS-CoV-2-infected and *n* = 3 uninfected (negative control). An overview of the experimental study design and timeline is presented in Appendix A. For each phase, cats were subdivided based on the route of inoculation. A total of *n* = 12 cats (*n* = 6 per phase) were inoculated by both intratracheal and intranasal routes (**IT + IN**), while *n* = 12 cats (*n* = 6 per phase) were inoculated by the intratracheal route alone (**IT Only**). A total of *n* = 6 cats (*n* = 3 per phase) were included as negative controls and were inoculated with sterile media by intratracheal and intranasal routes. Prior to inoculation, cats were anesthetized with ketamine (4 mg/kg), dexmedetomidine (20 µg/kg), and butorphanol (0.4 mg/kg) intramuscularly as previously described [26]. For intratracheal inoculation, IT Only (*n* = 12) and IT + IN (*n* = 12) cats were positioned in ventral recumbency and intubated as previously described [26,35], with the end of the endotracheal tube positioned within the distal trachea cranial to the tracheal bifurcation. The concentration of SARS-CoV-2, Isolate hCoV-19/USA/PHC658/2021 (Delta Variant) was normalized to 0.8 × 10^5^ TCID_50_ per ml in Dulbecco’s Eagle Modified Medium (DMEM, Gibco, Carlsbad, CA, USA), and a 3-cc syringe was utilized to intratracheally inoculate each cat with 3.2 × 10^4^ TCID_50_ SARS-CoV-2 per kg of body weight (0.8–1.7 × 10^5^ TCID_50_ total dose) followed by 2 mL of air from an empty syringe. A subset of cats (IT+IN) (*n* = 12 total, *n* = 6 per phase) was also intranasally inoculated with 0.4 × 10^4^ TCID_50_ SARS-CoV-2 (0.2 × 10^4^ TCID_50_ per nostril) similar to previously described methods [23]. Following inoculation, virus back-titration was immediately performed on Vero (CCL-81) cells, confirming that all cats received 0.4 × 10^4^ TCID_50_ SARS-CoV-2 intranasally and/or 3.2 × 10^4^ TCID_50_ SARS-CoV-2 per kg of body weight intratracheally. The remaining negative control cats (*n* = 6) were intranasally and intratracheally sham inoculated using equivalent volumes of sterile media (DMEM).

### 2.4. Clinical Evaluation

Animals were monitored twice daily for the development of SARS-CoV-2-specific clinical signs and evidence of morbidity by a licensed veterinary practitioner. Complete clinical evaluation was performed for all time points using a previously described clinical scoring system [26] that was additionally modified to include evaluation of gastrointestinal signs (e.g., diarrhea) in addition to body weight, body temperature, activity levels, respiratory effort, behavior, presence of ocular/nasal discharge, and evidence of coughing or wheezing. Similar to previous studies [26] each factor was assigned a score of 0 (normal), 1 (mild), 2 (moderate), and 3 (severe) as described in Table 1. Clinical parameters were then summated to assign individual animals a total clinical score every 24 h for the duration of the study (0 through 27). Cats were observed at rest for respiration rates, activity levels, and other notable clinical signs before stimulation. Oxygen saturation (SpO_2_) was measured using pulse oximetry at a stimulated state.

### 2.5. Sample Collection

Blood and nasal swab samples were collected under light sedation as previously described [26] from all Phase 1 cats (*n* = 15) at 0, 2, and 4, days post-inoculation (dpi), and from all Phase 2 cats (*n* = 15) at 0, 2, 4, 8, and 12 dpi. Blood and nasal swab samples collected at day 0 were used as baseline measurements for downstream analyses. Blood samples (up to 6 mL) were obtained from all cats via jugular venipuncture and immediately processed for flow cytometry and viral quantification. Nasal swabs samples were collected from both nares on all the time points mentioned earlier, placed in 2 mL tubes containing 200 µL phosphate-buffered saline (PBS), and stored at −80 °C for processing. At 4 dpi (Phase 1) and 12 dpi (Phase 2), a subset of SARS-CoV-2 infected cats (*n* = 12 per time point) and sham-inoculated cats (*n* = 3 per time point) were anesthetized for blood and nasal swab collection and then humanely euthanized (pentobarbital >80 mg/kg) and necropsied to collect tissue samples as previously described [26].

### 2.6. Virus Isolation

To estimate the infectivity of virus recovered from cats, we performed cell culture of nasal swabs collected at all timepoints throughout the study. Vero cells (CCL-81) were used to culture the nasal swabs collected at the abovementioned time-points and assess for cytopathic effects (CPE). Briefly, nasal swab samples stored in PBS-filled tubes were thawed for usage, vortexed vigorously for 20–30 s, and then inverted the swabs and centrifuged at 1500 rpm for 5 min. The swabs were discarded, and the remaining PBS served as the nasal swab inoculate. Prior to inoculation, Vero cells were plated in 24 well plates at 5 × 10^5^ cells/mL and cultured in growth media overnight at 37 °C in 5% CO_2_. For each sample, nasal swab inoculates of 25 µL were cultured in duplicate along with positive controls (25 µL viral stock- SARS CoV-2 isolate hCoV-19/USA/PHC658/2021 (Delta Variant)) and negative controls (25 µL Vero cell media). After adding samples, plates were incubated for 4–5 days at 37 °C in 5% CO_2_. We observed viral culture plates at 4 days post-inoculation with an inverted light microscope to document the presence of CPE. A positive result was defined by the presence of CPE at day 4 post-inoculation, and a negative result was characterized by having no CPE.

### 2.7. Histopathology

Necropsy was performed on all Phase 1 cats (*n* = 15 total) at 4 dpi (*n* = 12 SARS-CoV-2-infected cats and *n* = 3 uninfected cats), and at 12 dpi for all Phase 2 cats (*n* = 15 total; *n* = 12 SARS-CoV-2-infected cats and *n* = 3 uninfected cats). As outlined in previous studies [26], necropsy tissues were bisected and then placed into either 1 mL tubes and frozen at 80 °C, or into standard tissue cassettes that were then fixed in 10% neutral-buffered formalin for 96 h prior transferring to 70% ethanol for 72 h. Tissues were then trimmed and processed for histopathology as previously described [26,36]. Paraffin-embedded sections were trimmed to 5 µm, collected onto charged slides, and stained with hematoxylin and eosin (H&E) for microscopic evaluation by light microscope. Necropsy tissues were histologically evaluated for pathologic lesions reported in human COVID-19 patients [37,38,39,40] and in previous studies of SARS-CoV-2 infected cats [26]. Lung tissues, in particular, were evaluated for alveolar damage (e.g., pneumocyte necrosis, hyaline membrane formation) serous exudate/edema, alveolar fibrin deposition, alveolar histiocytes, perivascular infiltrates, type II pneumocyte hyperplasia, peri-bronchial inflammation, smooth muscle hyperplasia, thrombosis, and fibrinoid vasculitis. Masson’s trichrome stain was also performed on all lung tissues (*n* = 30) and evaluated for collagen deposition associated with inflammation. Heart tissues were evaluated for evidence of lymphocytic myocarditis, macrophage infiltration, small vessel thrombosis, myocardial disarray, and ischemic necrosis (infarction). Tracheal tissues were evaluated for evidence of ulceration, diphtheritic membrane formation, mucosal inflammation, submucosal gland inflammation, and muscular/serosal inflammation. Tracheobronchial lymph node (TBLN) tissues were evaluated for lymphoid hyperplasia, sinusoidal histiocytosis, sinusoidal neutrophil infiltration, edema, and perinodal inflammation. Nasal turbinate tissues were evaluated for ulceration, goblet cell hyperplasia, mucosal and submucosal inflammation. All tissues were assigned a quantitative histopathological score based on previously documented criteria [26,36,41]: 0 = no apparent pathology/change; 1 = minimal change (minimally increased numbers of inflammatory cells); 2 = mild change (mild inflammatory infiltrates, alveolar damage/necrosis, fibrin deposition and/or exudation); 3 = moderate change (as previously described, but more moderately extensive); 4 = marked changes (as previously described, but with severe inflammation, alveolar damage, hyaline membrane formation, necrosis, exudation, vasculitis and/or thrombosis). For quantitative scoring of fibrosis by Masson’s trichrome stain, tissues were assigned scores based on the following criteria: 0 = No increased collagen and/or fibrosis; 1 = Minimal increase in collagen associated with inflammation; 2 = Mild increase in collagen associated with inflammation; 3 = Moderate increase in collagen associated with inflammation; 4 = Severe fibrosis and marked increase in collagen associated with inflammation. All tissues were evaluated and scored by a board-certified veterinary pathologist blinded to study groups primarily to eliminate bias and to ensure scientific rigor.

### 2.8. Immunofluorescence Assay (IFA)

Paraffin embedded tissues from the right cranial lung of each cat were sectioned and fixed onto positively charged slides as previously described [26]. The tissues were deparaffinized through progressive processing steps involving toluene and ethanol. Following initial rehydration, slides were rinsed with deionized water for 10 min. Antigen retrieval was performed using a citrate-based antigen unmasking solution concentrate with a low-pH (H-3300, Vector Labs, Newark, CA, USA). Slides were placed in solution containing unmasking concentrate, brought to a boil, and left to cool on the benchtop for 30 min. Slides were then removed, dried and outlined with a PAP pen. Tissues were incubated overnight at 4 °C with a mouse monoclonal antibody to SARS-CoV-2 Nucleocapsid Protein (E8R1L, Cell Signaling Technology, Danvers, MA, USA) diluted 1:1000 in 10% normal goat serum (NGS). Following a 15 min wash with TBS (tris-buffered saline) solution, tissues were blocked with NGS for 1 h at room temperature, and then incubated for 30 min with a goat anti-mouse secondary antibody tagged with Alexa Flour 647 (4410S, Cell Signaling Technology, Danvers, MA, USA) diluted to 1:1000 in NGS. After subsequent washes, tissues were incubated with a rabbit polyclonal anti-ACE2 antibody (ab15348, Abcam, Waltham, Boston, USA) diluted 1:500 in NGS for 2 h at room temperature. Slides were then washed with TBS and incubated for 30 min at 20 °C with a goat anti-rabbit antibody conjugated with Alexa Flour 555 diluted 1:1000 in NGS. Following a final rinse, DAPI (4′,6-diamidino-2-phenylindole) was applied to serve as a nuclear counterstain. Slides were mounted using Shandon™ Immu-Mount (9990402, Thermo-Scientific), cover slipped, and visualized by confocal microscopy to assess the expression of target antigens. Each slide was examined at 4–20× with a Zeiss Axio Slide Scanner (Zeiss, White Plains, NY, USA).

### 2.9. Flow Cytometry

Blood was collected prior to inoculation to establish baseline values, then at each time point outlined above (Appendix A). The percentage of cells expressing CD4, CD8 and CD21 surface antigens was determined by incubating 50 µL of EDTA-treated blood with mouse monoclonal antibodies directed to the feline markers CD4 (Fisher, clone 3-4F4, FITC), CD8 (Southern Biotech, clone fCD8, PE), and CD21 (Bio-Rad, CA2.1D6, AF647) at the manufacturer’s recommended volume per test for 20 min in the dark at 4 °C. Red blood cells were lysed, and samples fixed using the TQ-Prep workstation and IMMUNOPREP reagents (Beckman Coulter Inc., Brea, CA, USA). Unstained and single stained controls were prepared for each experiment. Data were acquired using BD FACSDiva™ Software (Diva 9.0.1., San Jose, CA, USA) interfaced with a BD FACSAria™ SORP instrument (Becton Dickinson, San Jose, CA, USA). Data were analyzed using FlowJo 10.8.0. (Ashland, OR, USA). Compensation values were determined using single stained controls. Gating proceeded from singlets to lymphocytes to CD4, CD8, or CD21 markers. The percentage of lymphocytes positive for each marker was evaluated over time and compared to baseline values (0 dpi) and naïve control data to compare alterations in lymphocyte immunophenotype in response to SARS-CoV-2 infection.

### 2.10. Viral Genome Sequencing, Genome Assembly, Alignment and Phylogenetic Analysis

RNA was extracted from the viral inoculum and cranial lung and TBLN samples collected from cats at necropsy on both 4 dpi and 12 dpi. Following first-strand cDNA synthesis (Random hexamers and Superscript IV Reverse Transcriptase, Thermofisher, MA, USA), overlapping viral genome segments were generated using ARTIC V3 primers (https://github.com/artic-network/artic-ncov2019/tree/master/primer_schemes/nCoV-2019/V3, accessed on 5 May 2022). cDNA libraries were prepared using the Ligation Sequencing kit (SQK-LSK-109, Oxford Nanopore Technologies, Oxford, UK). DNA was cleaned for library preparation using Solid Phase Reversible Immobilization (SPRI) paramagnetic beads (Beckman Coulter, CA, USA). Following the manufacturer’s recommendations, the different samples were barcoded using Native Barcoding kit (EXP-NBD-104, Oxford Nanopore Technologies, Oxford, UK). Barcoded DNA libraries were then pooled and sequenced using GridION (Oxford Nanopore Technologies, UK) platform using a mean Q-score of 8.

Reference assembly was performed using the SARS-CoV-2 Delta variant (Isolate hCoV-19 /USA /PHC658 /2021). Samtools [42], minimap2 [43], and nanopolish [44] were used to obtain reference-assisted draft genomes. Viral genome ORF reader 4 (VIGOR4) was used for gene prediction in draft genomes using the curated libraries available online in the Virus Pathogen Resource (ViPR) [45] database. Open reading frames and individual genes (ORF 1a, ORF1ab, ORF3, ORF7, ORF8, ORF10, S, N, E, and M genes) identified from each draft genome were aligned to the reference genome (SARS-CoV-2 hCoV-19/USA/PHC658/2021) using MUSCLE aligner in MEGA-X [46].

### 2.11. Viral RNA Analysis

Viral RNA analysis was carried out on all necropsy tissues samples as follows. Tissue homogenates were made from samples collected from the cranial lung, nasal turbinates, distal trachea, kidney, olfactory bulb, TBLN, ileum, pancreas, and heart (left ventricle) of all cats involved in the study by adding the tissues to 600 µL of RLT (lysis buffer) in the tissue homogenizer. RNA was extracted from tissue homogenates using Qiagen RNeasy Mini kit (Qiagen, Germantown, MD, USA) according to the manufacturer’s recommendations and as previously described [26]. RNA from each sample was converted to cDNA using Superscript II reverse transcriptase (Invitrogen, Carlsbad, CA, USA) in individual reactions with random hexamers (Invitrogen) and treated with RNase Out (Invitrogen) prior to droplet digital PCR quantification as previously described [47]. Droplet Digital PCR [48] was performed to quantify SARS-CoV-2 viral loads using N1 and N2 primers (500 nM each) and probe (at 250) (Caltalog#10006713, Integrated DNA Technologies, Inc., Coralville, Iowa). The PCR reaction mixtures contained the following: 1.5 μL of N1, 1.5 μL of N2 probe, 10 μL of ddPCR Supermix for Probes (No dUTP) (Bio-Rad, Hercules, CA, USA) and 6.3 μL of cDNA template. Duplicate 20 μL samples were partitioned as previously described [26] using a QX200 droplet generator (Bio-Rad) and processed in a C1000 Touch Thermal Cycler (Bio-Rad) using the following protocol: 95 °C for 10 min for initial denaturation, 95 °C for 30 s, 55 °C for 60 min for annealing/extension for 45 cycles and 98 °C for 10 min for enzyme deactivation. For all the steps in this protocol, a ramp rate of 2 °C/s was used. A positive control (RNA extracted from SARS-CoV-2 Delta variant viral stock and diluted 1:15,000) and negative control (no template control, NTC) was included in each run as previously described [26]. Amplified samples were read in the FAM and HEX channels of a QX200 reader (Bio-Rad) and data were analyzed using Quanta soft Software 1.7 (Bio-Rad) and expressed as Log10 (copies/mL) [26].

### 2.12. Feline ACE-2 Analysis

Feline Angiotensin Converting enzyme (f-ACE2) RNA was quantified by ddPCR using methods and primers as previously described [26]. Total RNA from necropsy tissues was converted to cDNA and the concentration was normalized to 100 ng/µL. ddPCR reactions were prepared by adding 10 μL of Supermix for Probes (no dUTP) (Bio-Rad), 1 μL of primer/ probe mix and 9.45 μL of cDNA template. The following protocols were set up in the C1000 Thermal cycler to process the samples: 95 °C for 10 min for initial denaturation, 95 °C for 30 s, 58.8 °C for 60 min for annealing/extension for 44 cycles, and 98 °C for 10 min for enzyme deactivation. For all the steps in this protocol, a ramp rate of 2 °C/s was used. Amplified samples were read in the FAM channel of a QX200 reader (Bio-Rad), and data were analyzed as outlined above.

### 2.13. Lung Transcriptome Analysis

RNA Sequencing (RNA Seq) analysis was performed on lung tissues from all cats in this study (*n* = 30). RNA was extracted from lung tissues as outlined above, and cDNA library preparation and RNA sequencing were performed by Novogene Co Ltd. (Sacramento, CA, USA), generating paired-end reads at 150 bp length on an Illumina platform. Quality control of raw reads was conducted using Trim Galore wrapper scripts v0.6.5 (https://www.bioinformatics.babraham.ac.uk/projects/trim_galore/, accessed on 5 May 2022). Read ends with low-quality base calls (phred score ≤ 20) were trimmed in addition to adapter removal. Preprocessed fastq files were aligned to the *Felis_catus9.0* reference gene sequence file [49] using Bowtie2 (v2.2.1) [50] with default parameters and BAM files were transformed to sorted BAM format and quantified by SAMtools v1.6.0 [42]. The raw RNA expression results were integrated and processed in R (v4.1.1) computational environment and DESeq2 (v1.32.0) package [51] were used for library size normalization. To mitigate the within-group variance caused by outliers, we excluded 3 samples (8585, 6159, and 6515) due to their weak correlation with other individuals (Appendix A). Genes’ differential expression analysis was conducted using negative binomial distribution-based generalized linear model in DESeq2 package, and differentially expressed genes (DEGs) were determined with Benjamini–Hochberg (BH) adjusted *p*-value < 0.05. To investigate functions of DEGs between 4 dpi and uninfected individuals, and DEGs between two phases (4 and 12 dpi), hypergeometric test through DAVID [52] bioinformatics resources was performed on Gene Ontology (GO) and Kyoto Encyclopedia of Genes and Genomes (KEGG) pathway database. Expressed genes across all samples were used as backgrounds and only GO terms/KEGG pathways with BH adjusted *p*-value < 0.05 were retained in our enrichment analysis.

To determine correlation between gene expressions across samples, we employed WGCNA (v 1.70.3) to construct a gene co-expression network [53]. Five outlier samples were excluded (8585, 6159, 6515, 8836, and 7733) based on a hierarchical clustering analysis of all individuals (Appendix A). The top 60% variable genes (12402 out of 20670 expressed genes) were retained for gene clustering analysis. By using dynamic tree cut for branch cutting of gene dendrogram, different clustering modules marked by different colors were observed, and we merged modules whose expression profiles are very similar (correlation of their eigengenes >0.7). We correlated (Pearson’s correlation) eigengenes of 12 modules, defined as the first principle component of the corresponding expression matrix, with clinical and histological scores. Hub genes within significant modules were determined with the intra-modular connectivity is at top ten in each module.

### 2.14. Statistical Analysis

All the animals in the study were randomly assigned to each group. All cell culture experiments were conducted in duplicates. Results were analyzed with Graph Pad Prism 9.0 Software (La Jolla, CA, USA) and presented as the mean ± SEM when applicable. Kruskal–Wallis test, one-way-, and repeated measures ANOVA with Tukey post hoc analysis were used to compare differences in clinical disease score, histopathology, SARS-CoV-2 RNA, and feline ACE2 RNA between uninfected and among groups of SARS-CoV-2-infected individuals, between sample type, for each tissue individually, and between tissues. *p*-values less than 0.05 were considered as significant. Statistical methods for RNA Seq analysis data were performed using R-packages as described above in Section 2.13.

## 3. Results

### 3.1. SARS-CoV-2 (Delta Variant)-Infected Cats Develop Acute Clinical Respiratory Disease

Our previously described clinical scoring system for feline respiratory disease [26] was updated to include additional parameters such as oxygen saturation (SpO2) and more precise delineation of wheezing and coughing (Table 1). SARS-CoV-2 infected cats exhibited significantly increased overall clinical scores at 4, 5, and 12 days post-inoculation (dpi) when compared to sham-inoculated controls (Figure 1A). Clinical disease peaked at 4 and 5 dpi, then leveled off before the second increase in summated clinical scores was noted at 12 dpi. The most prominent clinical signs were lethargy, increased respiratory effort, and wheezing (Figure 1B). Throughout the 12-day study, wheezing was observed in 8 out of 12 SARS-CoV-2-infected cats. Lethargy and increased respiratory effort were noted in all 12 infected cats. Out of 12 infected cats, 6 displayed coughing, peaking on 4 and 8 dpi, and 5 of the 12 infected cats developed pyrexia (temperature ≥ 39.2 °C). All infected cats had altered behavior. None had notable ocular or nasal discharge or weight loss. Interestingly, 3 out of 12 infected cats developed diarrhea and/or hyperemesis later in the study, emerging at 10 to 12 dpi. Sham-inoculated cats did not exhibit any measurable clinical signs during the 12-day study.

In addition, methods and routes of infection influenced clinical signs in both Phase 1 (4-day) and Phase 2 (12-day) studies. The summated clinical scores per cat indicate that dual inoculation through both IT and IN routes resulted in elevated clinical scores compared to IT-only inoculation (Appendix A). Clinical scores were significantly elevated in dual (IT + IN) inoculation compared with IT-only inoculation at 4 dpi in both the 4-day and 12-day studies, with that significant difference persisting through day 12 of the 12-day study. Sham-inoculated cats were unaffected clinically by either route of inoculation.

### 3.2. Infectious Virus Is Shed via Nasal Secretions up to 4 Days Post-Inoculation

Infectious virus was recovered from all SARS-CoV-2 (Delta variant) infected cats (*n* = 24) in both Phase 1 and Phase 2 of this study. Cytopathic effects were detected in cell cultures inoculated with nasal swab samples collected at 2 dpi (*n* = 24) and 4 dpi (*n* = 24) as compared to CPE in positive control wells. However, nasal shedding of infectious virus particles appears to cease prior to 8 dpi as no CPE was detected in nasal swab samples collected from Phase 2 infected cats (*n* = 12) at 8 dpi or 12 dpi (Appendix A).

### 3.3. Sars-Cov-2 (B.1.617.2) Causes Severe Lung Pathology in Domestic Cats Akin to COVID-19

Lung tissues collected from all SARS-CoV-2-infected cats in Phase 1 (4 dpi; *n* = 12) and in Phase 2 cats (12 dpi, *n* = 12) were grossly examined and compared to those from sham-inoculated cats (*n* = 6) (Figure 2A–C). Compared to lungs from healthy sham-inoculated cats (Figure 2A), the lungs of Phase I cats infected with the Delta variant of SARS-CoV-2 exhibited large, multifocal to coalescing areas of dark red pulmonary consolidation, hemorrhage, and pulmonary edema at 4dpi: most often in a cranioventral distribution (Figure 2B). The lungs of Phase II SARS-CoV-2 infected cats exhibited lesions more consistent with an interstitial pattern at 12 dpi characterized by patchy consolidation throughout the lungs which were firm and failed to collapse (Figure 2C). TBLNs of all SARS-CoV-2-infected cats were diffusely enlarged to 4–5 times normal size in both Phase 1 (4 dpi, *n* = 12) and Phase 2 (8 dpi, *n* = 12). No significant difference in the degree or distribution of gross lesions were observed between different SARS-CoV-2 inoculation routes (IT Only vs. IT+IN) in either Phase of this study.

Microscopic evaluation of selected necropsy tissues (lung, trachea, nasal turbinates, TBLN, heart) was performed for all study animals in both Phase 1 and Phase 2 of the study. Tissue sections from all sham-inoculated animals (*n* = 6) were histologically unremarkable (Figure 2D and Appendix A). Histologic evaluation of lung tissues from SARS-CoV-2 (Delta variant) infected cats in Phase 1 (4 dpi) revealed distinct evidence of diffuse alveolar damage (DAD) characterized by marked alveolar histiocytosis (12/12 cats), discrete foci of alveolar inflammation and necrosis (10/12 cats), and disruption of the vascular architecture (vasculitis) by infiltrating neutrophils and lymphocytes (10/12 cats) (Figure 2E). Marked perivascular lymphocytic infiltrates (11/12 cats) and peribronchial inflammation (12/12 cats) were also observed at 4 dpi, and the interstitial space surrounding small and large caliber vessels was frequently expanded by large amounts of perivascular edema (10/12 cats), indicating acute vascular injury (Figure 2F). Numerous SARS-CoV-2 infected cats in Phase 1 (9/12) also exhibited necrotizing bronchopneumonia, occasional syncytial cells, and varying degrees of hyaline membrane formation (7/12 cats) that was most often associated with regions of hemorrhage and proteinaceous edema fluid present within the alveoli (Figure 2G). The degree of alveolar inflammation (7/12 cats) and perivascular inflammatory infiltrates (10/12 cats) was less severe in Phase 2 (12 dpi) compared to Phase 1 (4 dpi), although evidence of alveolar histiocytosis (12/12 cats), peribronchial inflammation, pulmonary edema (6/12 cats), and vasculitis (6/12 cats) were still apparent in SARS-CoV-2 infected cats at 12 dpi (Figure 2H). The trachea of SARS-CoV-2 (Delta variant) infected cats in Phase 1 (4 dpi) exhibited varying degrees of mucosal (10/12 cats) and submucosal (12/12 cats) effacement by inflammatory cells was multifocally ulcerated with necrosis and diphtheritic membrane formation in 4/12 cats (Figure 2I). Mucosal inflammation was much less severe in cats in Phase 2 (12 dpi), although all cats in Phase II exhibited some degree of chronic inflammation within submucosal glandular tissues (Figure 2I, inset), characterized by large coalescing aggregates of lymphocytes and macrophages often forming follicular structures.

Histopathologic lesions in all necropsy tissues revealed that different SARS-CoV-2 inoculation routes (IT Only vs. IT + IN) had no effect on the degree or distribution of histologic lesions in either phase of this study for all tissues evaluated (Appendix A). Histologic scores were consequently summated for quantitative analysis of lesions at each timepoint. Histologic scoring of lung lesions revealed significant pulmonary pathology in SARS-CoV-2 infected animals which was most pronounced at day 4 post-inoculation (Appendix A) compared to uninfected controls (*p* < 0.0001) and SARS-CoV-2 infected cats at 12 dpi (*p* < 0.0001). Total lung pathology was also significantly increased in SARS-CoV-2 infected cats at 12 dpi compared to uninfected cats (*p* < 0.0001). When looking at individual pathologic features, alveolar inflammation (*p* < 0.05), alveolar histiocytosis *p* < 0.01), and peribronchial inflammation (*p* < 0.0001) were significantly increased in cats at 4 dpi compared to uninfected cats, as well as perivascular inflammation (*p* < 0.0001) and vasculitis (*p* < 0.05) (Appendix A). Perivascular inflammation (*p* < 0.001) and peribronchial inflammation (*p* < 0.001) were also significantly elevated in SARS-CoV-2 infected cats at 4 dpi compared to 12 dpi. Overall, total lung pathology was more severe in Delta variant-infected cats at 4 dpi compared to 12 dpi (*p* < 0.0001) and uninfected controls (*p* < 0.0001) (Appendix A). Histologic scoring of Masson’s trichome stained tissues reveled significantly increased fibroblasts and collagen deposition in the lungs of SARS-CoV-2-infected cats at both 4 dpi (*p* < 0.01) and 12 dpi (*p* < 0.01) compared to sham-inoculated cats (Figure 3 and Appendix A).

Mucosal inflammation in the trachea was significantly elevated in SARS-CoV-2 infected at 4 dpi compared to uninfected controls (*p* < 0.0001) and SARS-CoV-2 infected cats at 12 dpi (*p* < 0.0001). Inflammation within the submucosa was also significantly elevated in SARS-CoV-2 infected cats at both 4 dpi (*p* < 0.0001) and 12 dpi (*p* < 0.05). Tracheal ulceration (*p* < 0.05) and diphtheritic membrane formation (*p* < 0.05) were also significantly elevated in SARS-CoV-2 infected cats at 4 dpi compared to 12 dpi (Appendix A). SARS-CoV-2-infected animals in Phase II exhibited a significantly increased degree of lymphoid hyperplasia in the TBLN at 12 dpi compared to uninfected controls (*p* < 0.0001) and SARS-CoV-2 infected cats at 4 dpi (*p* < 0.0001), as well as increased sinusoidal histiocytosis compared to 4 dpi (*p* < 0.05) and perinodal inflammation compared to both uninfected controls (*p* < 0.05) (Appendix A). Perinodal inflammation was also observed in SARS-CoV-2 infected cats at 4 dpi (*p* < 0.01) in the TBLN. No significant histopathologic findings were observed in heart tissues or nasal turbinates at either time point.

### 3.4. Sars-Cov-2 Antigen Is Detected in ACE2-Expressing Lungs of Domestic Cats during Infection with the Delta Variant

Immunofluorescence assay (IFA) was performed on lung samples from all sham-inoculated (uninfected) and SARS-CoV-2 infected cats to detect SARS-CoV-2 positive cells in concert with ACE2 expression. SARS-CoV-2 antigen was detected in a large proportion of ACE2-expressing epithelial cells (pneumocytes, peribronchiolar glandular epithelium), immune cells (alveolar macrophages and lymphocytes), and vascular endothelial cells of all SARS-CoV-2 infected cats at both 4 dpi (12/12 cats) and 12 dpi (12/12 cats) (Figure 4). Quantification of IFA fluorescent intensity revealed that the presence of viral antigen was significantly increased in SARS-CoV-2 infected cats at 4dpi when compared to uninfected controls (*p* < 0.0001) and SARS-CoV-2 infected cats at 12 dpi (*p* < 0.01) (Appendix A), and was also significantly increased at 12 dpi (*p* < 0.0001) compared to uninfected controls.

### 3.5. Domestic Cats Exhibit Divergent Lymphocyte Immunophenotypes during SARS-CoV-2 Infection

Flow cytometry was used to evaluate changes in the percentage of blood lymphocytes positive for CD4, CD8, and CD21 surface antigens over time in SARS-CoV-2-infected and sham-inoculated (uninfected) cats. Changes in SARS-CoV-2-infected cats were compared to baseline values and data from naïve control cat samples to compare alterations in lymphocyte immunophenotype in the presence of SARS-CoV-2 infection. No significant differences in immunophenotype were observed between treatment groups in SARS CoV-2 infected cats (IT Only vs. IT + IN) at any time point in either phase of the study (4 dpi and 12 dpi), so data were summated for both Phase 1 and Phase 2 and presented in Figure 5.

No significant differences were observed in immunophenotype (CD4, CD8, CD21, CD4:CD8) for sham inoculated (uninfected) cats (Figure 5A). In both Phase 1 and Phase 2, SARS-CoV-2-infected cats exhibited a significant decrease in the proportion of CD21+ cells at 2 dpi (*p* < 0.0001) and 4 dpi (*p* < 0.0001), mirrored by a simultaneous increase in the proportion of CD4+ cells (2 dpi *p* < 0.0001; 4 dpi *p* < 0.0001) (Figure 5B,C). This decrease in CD21+ cells extended through 8 dpi (*p* < 0.001) and 12 dpi (*p* < 0.01) of Phase 2 of the study, whereas the increase in CD4+ cells extended to 8 dpi (*p* < 0.05) but not at 12 dpi. No significant differences were observed in the proportion of CD8+ cells or the CD4:CD8 ratio in SARS-CoV-2 infected cats at any timepoint in Phase 1 or Phase 2 of the study.

### 3.6. Viral Sequencing and Genome Assembly

The viral genomes sequenced from the three samples (Viral RNA, lung, and lymph node), the total number of reads generated for each genome, and the approximate coverage for the genomes are listed in Table 2.

Individual genes were aligned to the reference genome (hCoV-19/USA/PHC658/2021). A synonymous mutation was noted at nucleotide position 13131 (Amino acid position-4377) in ORF 1ab of the viral genome derived from the lungs (Table 3).

Besides the synonymous mutation in the lung, no other mutations were noted in viral genomes from lungs and tracheobronchial lymph nodes compared with the sequenced genome of inoculated viral component used in this study or the reference genome.

### 3.7. Viral RNA and ACE-2 Expression in Tissues during SARS CoV-2 (Delta Variant) Infection

SARS-CoV-2 viral RNA and f-ACE2 RNA expression were quantified in lungs nasal turbinates, TBLN, ileum, olfactory bulb, kidney, distal trachea, and left heart ventricle, of all SARS-CoV-2 infected cats (*n* = 24) and sham-inoculated control cats (*n* = 6) using ddPCR as outlined above. No significant differences in SARS-CoV-2 viral load or f-ACE2 expression were observed between SARS-CoV-2 infected cats at 4 dpi or 12 dpi when evaluated by inoculation route (IN + IT vs. IT Only). Data for necropsy tissues collected at these time points were thus combined and analyzed as complete data sets for each timepoint collected (4 dpi vs. 12 dpi). Complete data sets for individual inoculation routes are presented in the Appendix A (Appendix A)

Viral RNA was detected in all cats and in all necropsy tissues evaluated at 4 dpi and at 12 dpi (Figure 6A), with the exception of 3/12 cats which had no measurable virus in the lung at 12 dpi. SARS CoV-2 RNA was not detected in any of the sham-inoculated cats at 4 dpi or 12 dpi (Appendix A). SARS-CoV-2 viral copies were significantly higher at 4 dpi in the distal trachea (*p* ≤ 0.0001), nasal turbinates (*p* = 0.0011), and kidney (*p* = 0.0126) compared to both control and SARS-CoV-2 infected tissues collected on 12 dpi. SARS-CoV-2 viral load in the olfactory bulb was also higher at 4 dpi compared to 12 dpi, but this trend was not significant (*p* = 0.08) (Appendix A).

Analysis of feline ACE2 expression in necropsy tissues (Figure 6B) revealed that f-ACE2 expression in the distal trachea was significantly decreased at 4 dpi compared to sham-inoculated (uninfected) controls. Overall, there appeared to be reduced expression of f-ACE2 in the lung and heart of CoV-2 infected animals at 12 dpi compared to control animals, but this was not statistically significant. No other significant changes were observed in ACE2 RNA expression.

### 3.8. Inflammatory Pathways Are Dysregulated in Acute Phase and Neurological Pathways Are Altered in Chronic Phase with a Shift in Transcriptomic Profile from One Phase to Another

In order to investigate whether SARS-CoV-2 infection alters gene regulation in transcriptome level, RNA-seq analysis of the lung samples was conducted to determine differentially expressed genes (DEGs) and corresponding pathway and gene ontology (GO) enrichment. In total, SARS-CoV-2 infection in Phase 1 showed 576 upregulated and 141 downregulated genes at 4 dpi compared to sham-inoculated (uninfected) control cats (Appendix A). When comparing DEGs from uninfected controls to infected cats at 4 dpi, there is upregulation of many genes associated with activation of innate immunity and SARS CoV-2 disease severity such as KIF11, IRF7, OAS1, BATF2, ENSFCAG00000003942, ENSFCAG00000001885, IF16, and BPIFA1. KEGG pathway analysis of these DEGs identified gene pathways that function in the acute cellular response to external antigens including cell cycle, antigen processing and presentation and receptor interaction pathways (Figure 7A). GO terms were enriched in biological process (BP), cellular component (CC) and molecular function (MF) sections (Figure 7B), and they also reflected upregulation of genes such as GINS4, CDCA8, and AURKB-Aurora B, the latter of which is essential for regular cell cycle and proper chromosome segregation [54].

We also investigated differences in gene expression between infected cats in Phase 1 (4 dpi) and Phase 2 (12 dpi) in order to evaluate transcriptome dynamics during the recovery phase of infection. Overall, there were 789 upregulated and 1466 downregulated genes in lung samples collected at 12 dpi compared to 4 dpi (Appendix A). Interestingly, several genes involved in neurodegenerative diseases, such as PATZ1, SOX-4. GPRC5B, and ZNF423, were upregulated in SARS-CoV-2 infected cats at 12 dpi compared to earlier time points (4 dpi). In addition to the upregulated pathways observed during acute CoV-2 infection (4 dpi), KEGG Pathway analysis identified DEGs during the recovery phase (12 dpi) were also enriched in many neurodegenerative diseases (Figure 7C), and GO terms were enriched in protein catabolic process and endopeptidase activity (Figure 7D). There were also a small number of DEGs between 12 dpi and uninfected control lung samples (28 DEGs in total, Appendix A) suggesting a homogeneity of gene regulation between cats recovering from SARS-CoV-2 infection (at 12 dpi) and uninfected cats.

We next employed weighted correlation network analysis (WGCNA) on the lung RNA-Seq data in Phase 1 (4 dpi) and Phase 2 (12 dpi), and identified 12 gene modules (11 modules of genes with high absolute intra-modular correlations and 1 module with the remaining uncorrelated genes) (Appendix A) with the number of genes ranging from 323 (Cyan module) to 2765 (Brown module) (Figure 8A). We then identified modules significantly associated with histological and clinical scores. There were significant positive correlations (*p* < 0.05) between several bio-indicators (reduced activity/lethargy, coughing, increased respiratory effort, weight change, intra-alveolar inflammation/fibrin deposition, alveolar histiocytosis, perivascular inflammation, perivascular edema, vasculitis, and peribronchial inflammation) and the expression profile of genes in the black module. In contrast, genes in the purple module were negatively correlated with the majority of bio-indicators identified in the black module (Figure 8A). Interestingly, genes clustered in the black module were enriched in cell cycle, viral infection, and neurological disease pathways, while genes clustered in the purple module were related to RNA processing pathways (Appendix A), highlighting the tight causation of these pathways with corresponding histopathology after SARS-CoV-2 infection. Based on scale free network topologies, hub genes were characterized by more numbers of highly correlated connections with downstream targets [52]. As a result, we identified the top 10 hub genes in both the black and purple modules which exhibit high connectivity to other intra-modular genes (Figure 8B) and may serve as upstream regulators of related clinical symptoms after viral infection. The hub genes identified in the black modules are—ASF1B, AURKB, TYMS, HDGF, CDCA8, EIF4A1, CCT3, CALR, and RRM2, whereas the hub genes found in purple modules are NKTR, RBM5, SRRM2, OGT, DDX5, PNISR, ZNF331, ENSFCAG00000027050, ENSFCAG00000030565, and FNB. In addition to these hub genes, other intra-modular genes involved in inflammatory and immune modulatory pathways were found including MAP4K, ILF-2, NFKBID, ICAM1, and MASP2.

## 4. Discussion

The results from this study not only strengthen the utility and translational adaptability of the domestic cat model to study acute COVID-19 and SARS-CoV-2 variants of concern, but also provide insight into the mechanisms driving the inflammatory response to infection. First, domestic cats infected with only 3.2 × 10^4^ TCID_50_ SARS-CoV-2 per kg of body weight of the Delta variant exhibited clinical signs and pulmonary lesions that align closely with hospitalized COVID-19 patients. This includes clear evidence of diffuse alveolar damage in infected cats’ lungs paired with marked clinical signs consisting of lethargy, increased respiratory effort, and coughing or wheezing. The clinical signs and histopathology were similar to that seen in our previous study, which utilized wildtype SARS-CoV-2 at about 10 times the dose of the Delta variant used in this study [26]. Based on these findings, the Delta variant of SARS-CoV-2 is significantly more virulent than the original lineages in the domestic cat, similar to the increased virulence of the Delta variance in humans [55].

Inoculating domestic cats via both the intranasal and intratracheal routes resulted in enhanced clinical scores versus intratracheal inoculation alone, but the route of inoculation had no other significant effect on pulmonary pathology, viral load, or transcriptome profile of infected cats at two time points studied. While these findings suggest a potentially important role of nasal infection in the development of clinical disease, this finding could also be due to increased dosing in these cats, and is generally not supported by evidence of increased pathology or viral replication with the addition of the intranasal route.

Several differences in clinical disease were noted between wildtype SARS-CoV-2 and Delta variant infection in the domestic cat. Wheezing and coughing were frequently noted clinical signs in infected cats, and these clinical signs tended to be more readily apparent in cats infected with the Delta variant compared to wildtype SARS-CoV-2 [26]. Clinical manifestations of experimental infection were also consistent with symptoms of upper respiratory disease observed in cats naturally infected with the Delta variant [56], and gastrointestinal signs were also observed in 3/12 cats on or after 10 dpi, reflecting similarities in clinical symptoms of humans with Delta infection [57]. While the features of diffuse alveolar damage observed on histology in all infected cats at 4 dpi is similar to our previous studies using wildtype SARS-CoV-2 [26], we also identified evidence of vasculitis and marked peri-bronchial and peri-vascular edema at 4 dpi, suggesting acute vascular injury is a prominent pathologic feature of Delta variant infection in cats as it is in humans. The hyaline membrane formation in the lungs and diphtheritic membrane formation in trachea observed in infected cats at 4 dpi are also important features of human COVID-19 patients, and the fact that total lung pathology was more severe in Delta variant-infected cats correlates well with the increased pathogenicity of the Delta variant in people [58]. Importantly, histological signs of acute inflammation are reduced by 12 dpi, but are replaced by early indicators of chronic pulmonary inflammation, such as organization and fibrosis (Figure 3) which may provide the foundation for a translational animal model to study the protracted consequences of long COVID.

SARS-CoV-2 virus was detected in all infected cat tissues, including the olfactory bulb, with a significantly higher viral loads in several tissues at 4 dpi compared with 12 dpi, suggesting that peak viral load occurs in the acute phase of infection as in humans [59,60]. Interestingly, f-ACE2 expression is reduced in the trachea after infection with Delta variant. While further studies would be required to elucidate this mechanism, this could be due to SARS-CoV-2 viral mediated endocytosis of the ACE2 receptors that results in reduced expression [61].

Our study also identified significant shifts in systemic lymphocyte immunophenotypes during SARS-CoV-2 infection. The increased proportion of CD4+ T cells through 8 dpi may be associated with a T cell-mediated proinflammatory (Th1) response as it is directly proportional to disease severity, although this cannot be concluded without additional analyses [62,63]. Interestingly, the proportion of B cells was decreased in SARS-CoV-2 infected cats throughout the course of this study, even though many studies indicate that the frequency of B cells is increased in severe COVID-19 patients [64]. However, other studies in untreated patients with COVID-19 pneumonia reported decreased numbers of total and naïve B cells, along with decreased percentages and numbers of memory switched and unswitched B cells [65], likely due to the recruitment and exhaustion of partially mature B cells as a result of exaggerated immune activation. While CD8+ T cells are essential for viral clearance during acute viral infections in response to most respiratory viruses, we did not appreciate an increase in CD8+ T cell frequency or CD4:CD8 ratio [66]. Although further studies are needed to elucidate the protective mechanism of CD8+ T cells and other lymphocyte subtypes in viral clearance in domestic cats, these results provide insight and similarities of immune cell discrepancies in cats compared to humans with COVID-19.

An important outcome of this study was the bioinformatics analysis of gene expression from lung tissue specimens, which delineated broad transcriptomic profiles of lung tissues in healthy, uninfected cats compared to SARS CoV-2 infected cats at different timepoints of infection (4 dpi and 12 dpi). Our results show significant alterations in DEGs of cats with SARS CoV-2 that are comparable to studies in humans and other animal models of COVID-19, providing an additional resource to understand the pathogenesis and strategies to mitigate the disease.

Upregulation of many genes associated with activation of innate immunity and SARS CoV-2 disease severity were identified during acute (4 dpi) SARS-CoV-2 (Delta variant) infection in cats, such as KIF11, IRF7, OAS1, BATF2, IF16, and BPIFA1. KIF 11 encodes a motor protein that belongs to the kinesin-like protein family, and inhibition of this gene has been shown to attenuate Influenza virus-mediated cytopathic effect and decrease viral replication [67]. In fact, KIF11 inhibitors have been used to study their impact on viral replication and as a therapeutic agent against influenza viruses, thus highlighting a potential target for SARS CoV-2 infection. Upregulation of IRF7, OAS1, and BATF2 was previously documented in the human lung after the first 48 h of SARS CoV-2 infection [68], and dysregulation of these genes may provide clues to the function of acute cellular responses to external viral antigens in relation to antigen processing and receptor interaction pathways. A recent study in COVID-19 host genetics found that BATF2 controls the production of cytokines from CD4+ T-cells and macrophages in mice [69] and participates in interferon signaling [70], highlighting a potential immunotherapeutic target to mitigate COVID-19. KEGG pathway analysis also identified upregulation of cytokine–cytokine interaction and antigen presentation pathways during acute Delta variant infection (4 dpi), which are important pathways in restricting viral infection [71]. Differences in these pathways have been documented as a predominant pathway that regulates other pathways in earlier COVID-19 studies [72], and may be further investigated in future studies to identify additional therapeutic targets.

An interesting approach to evaluate the relationships of gene expression profiles with clinical and pathologic features of disease is weighted correlation network analysis (WGCNA). WGCNA is a bioinformatics approach that can be used to find clusters (modules) of highly correlated genes and to relate these clusters to one another and to external sample traits [53]. By employing WGCNA on lung samples from infected cats at both 4 dpi and 12 dpi, we identified 11 gene modules with high absolute intra-modular correlations—a few of which were significantly associated with histological and clinical scores (black and purple modules). These modules contain specific “hub” genes, that maybe upstream regulators or co-factors regulating any number of interconnected downstream genes in associated pathways or processes. Interestingly, most of the hub genes in our study have been linked to severe SARS-CoV-2 infection in humans, and elucidation of these gene pathways in cats may provide exploitable pathways to understand COVID-19 pathogenesis and improve therapeutic efficacy. For example, AURKB (a gene encoding Aurora B) is dysregulated during acute SARS-CoV-2 infection and highly associated with clinical signs and lung pathology in cats. AURKB controls cell cycle progression and proper chromosome segregation, and may play a role in SARS CoV-2 nucleocapsid phosphoprotein (N-protein) mutation since important Aurora kinase B phosphorylation sites (R203 and G204) are commonly mutated residues of N-protein [54,73]. TYMS is involved in the CD80/86 proinflammatory axis, and is upregulated in severe COVID-19 patients [74]. Dysregulation of this gene pathway may contribute to downstream effects of antigen presentation and sustained T cell activation [75]. Lastly, the 5′ cap of SARS CoV-2 is recognized by Eukaryotic initiation factor 4A-1 (EIF4A1) [76] and is known to be significantly increased in the platelet proteome of COVID-19 patients thought to affect viral replication and downstream activation of platelets, resulting in thrombotic microangiopathy and increased mortality [77]. Collectively, the results of our WGCNA analysis identified a substantial network of dysregulated genes with many downstream effects that are linked to clinicopathologic data. Targeting these “hub genes” (genes with similar co-expression patterns and multiple roles during infection) directly or through a combination with other inter-modular genes may provide a novel pathway to improve therapeutic intervention or develop prognostic biomarkers to diagnose infection due to the broad implications these genes have on downstream mediators of the acute inflammatory response.

While our study as a whole focuses on the adaptability and usefulness of the domestic cat as an animal model for COVID-19, it is also critical to understand the zoonotic potential for the domestic cat. Sequencing indicated one ORF mutation between the SARS-CoV-2 inoculum and SARS-CoV-2 collected from infected cat lung. Further investigation is needed to clarify the significance of that finding. Additionally, nasal swabs collected at 2 and 4 dpi contained infective virus but were no longer infective by day 8. This viral clearance from nasal tissues can likely be attributed to clearance of infection and further be supported by the similarity of gene expression between healthy and infected cats by 12 dpi, indicating recovery from infection. These findings highlight a crucial need to address zoonotic potential in the acute phase of SARS-CoV-2 infection in the cat, especially in regard to new emerging viral variants. Domestic cats are often in close proximity with their owners and could be a risk for transmission or act as an animal reservoir for continued mutation and variance emergence in the future.

In summary, this study focuses on the acute phase of SARS-CoV-2 (Delta variant) infection in domestic cats. There continues to be a shortage of animal models for chronic, long-term COVID-19, and the domestic cat model may provide an advantageous model to evaluate chronic disease pathology and long-term effects beyond day 12 of infection (i.e., Long COVID). This model is also highly adaptable to comorbidity studies as cats are prone to several diseases linked to increased severity in COVID-19, such as obesity, diabetes mellitus, hypertension, and renal disease. Limitations to this study include the sample size and limited time points of study at 4- and 12-days post inoculation for many measurements. It is clear from our findings that significant changes occur between days 4 and 12, which highlight a need to better elucidate the mechanisms resulting in those changes. In conclusion, the domestic cat model continues to be an adaptable and useful translational model for acute COVID-19, even as new variants emerge. Clinical and pathologic findings mimic human disease, and the inflammatory responses are also similar to that seen in patients hospitalized with COVID-19. Identification of key gene hubs or intermodular genes that are dysregulated by SARS-CoV-2 infection offer potential targets for therapeutics or prognostic biomarkers for SARS-CoV-2 infection. Based on these findings, cats should continue to be an important animal resource for COVID-19 studies and therapeutic and preventive trials.

## Figures and Tables

**Figure 1 viruses-14-01207-f001:**
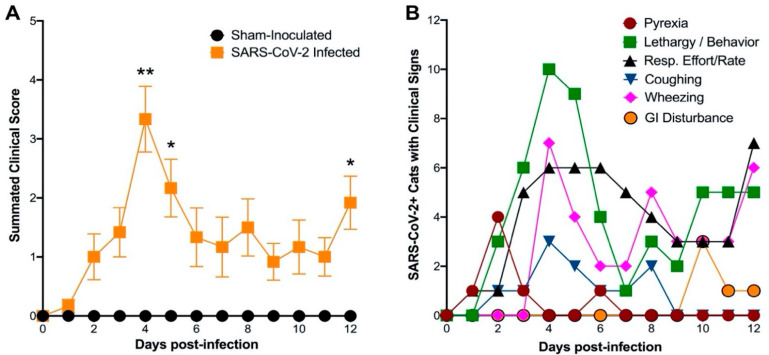
SARS-CoV-2 (Delta variant) infection results in clinical disease in domestic cats. Clinical parameters were assessed using a modified feline respiratory disease clinical scoring system (Table 1). (**A**) Clinical parameters were summated to provide an overall clinical score per cat per day. Similar to previous studies [26], clinical severity peaked at 4 dpi and significantly higher than sham-inoculated cats at 4 dpi (*p* = 0.0012), 5 dpi (*p* = 0.0132), and 12 dpi (*p* = 0.0178). (**B**) The number of cats displaying clinical signs is shown (*n* = 12). Lethargy, increased respiratory effort, and wheezing were the most prominent clinical signs and peaked on 4 and 5 dpi. Pyrexia peaked on 2 dpi. Pyrexia was noted in 5/12 cats, coughing in 6/12, and wheezing in 8/12 cats during the 12-day study window. GI disturbance was noticed in 3/12 cats, emerging at 10 dpi. Data are expressed as means ± SEM. Statistical comparisons were made via mixed effect analysis. * *p* < 0.05; ** *p* < 0.01.

**Figure 2 viruses-14-01207-f002:**
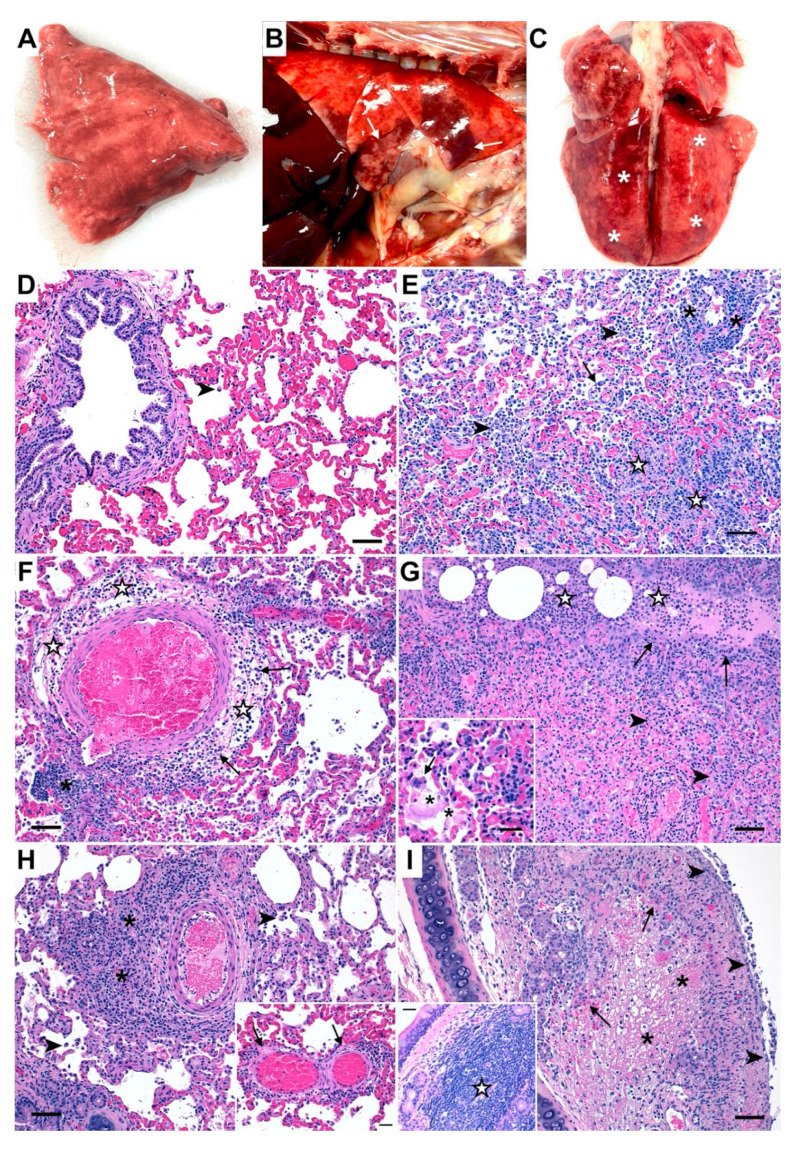
SARS-CoV-2 (Delta variant) infection causes severe pathologic lesions and acute alveolar damage in domestic cats. Compared to lungs from healthy sham-inoculated cats (**A**), the lungs of SARS-CoV-2 (Delta variant) infected cats were markedly consolidated at 4 dpi (**B**), with large, multifocal areas of hemorrhage and pulmonary edema that were most often in a cranioventral distribution (white arrows). At 12 dpi (**C**), the lungs of SARS-CoV-2 infected cats were firm and failed to collapse with patchy consolidation throughout (white asterisks). The lungs of healthy, sham-inoculated (uninfected) cats were histologically normal (**D**), with open bronchi/alveoli and minimal alveolar macrophages (arrowhead). At 4 dpi (**E**), lung tissues from SARS-CoV-2 (Delta variant) infected cats contained large numbers of alveolar histiocytosis (arrowheads), discrete foci of alveolar inflammation and necrosis (white stars), perivascular lymphocyte infiltrates (asterisks), and vasculitis (arrow). (**F**) Large amounts of perivascular edema (white star) were present at 4 dpi surrounding small and large caliber vessels, indicating acute vascular injury, and were accompanied by perivascular lymphoid aggregates (asterisk) and infiltration by neutrophils and macrophages (arrows). (**G**) In several cats, the bronchial epithelium was effaced (arrows) at 4 dpi and the bronchial lumen was filled with large numbers of neutrophils and macrophages (white stars) which frequently spilled out into the adjacent alveoli where they were accompanied by large amounts of necrotic cellular debris (arrowheads). Occasional syncytial cells (arrow) and varying degrees of hyaline membrane formation (asterisks) were observed in several cats at 4 dpi (**G inset**). The overall degree of alveolar histiocytosis (arrowheads) and perivascular inflammatory infiltrates (asterisks) was less severe at 12 dpi (**H**), although evidence of vasculitis (**H inset**, arrows) were still apparent in SARS-CoV-2 infected cats at this later timepoint. The trachea of SARS-CoV-2 (Delta variant) infected cats at 4 dpi (**I**) was multifocally ulcerated (arrowheads) and accompanied by submucosal necrosis (asterisks) and neutrophilic infiltrates (arrows). At 12 dpi (**I inset**) inflammation was less apparent at the tracheal mucosa, although submucosal glands were frequently expanded and effaced by large coalescing aggregates of lymphocytes and macrophages often forming follicular structures (white star). Magnification: (**D**–**I**) 20×, scale bar = 50 µm; (**G inset, H inset, I inset**)) 40×, scale bar = 25 µm.

**Figure 3 viruses-14-01207-f003:**
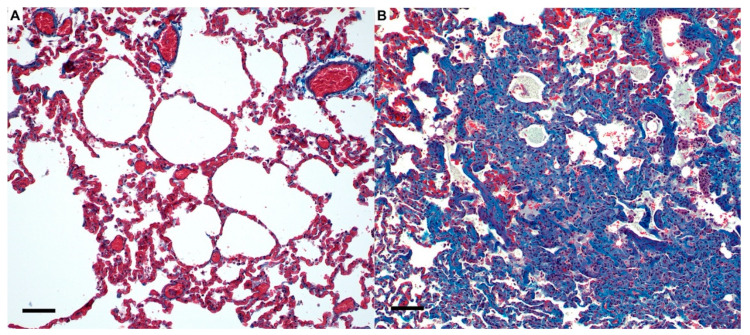
Masson’s trichrome stain of lungs during SARS-CoV-2 infection in domestic cats. (**A**) Normal lung from sham-inoculated cats exhibiting minimal amounts of collagen (blue) present within the interstitium and surrounding vascular structures. (**B**) In SARS-CoV-2 infected cats (12 dpi), alveolar septa are expanded by large amounts of collagen (blue) and proliferating fibroblasts that are strongly associated with areas of inflammation. Magnification 20×, scale bar = 50 µm.

**Figure 4 viruses-14-01207-f004:**
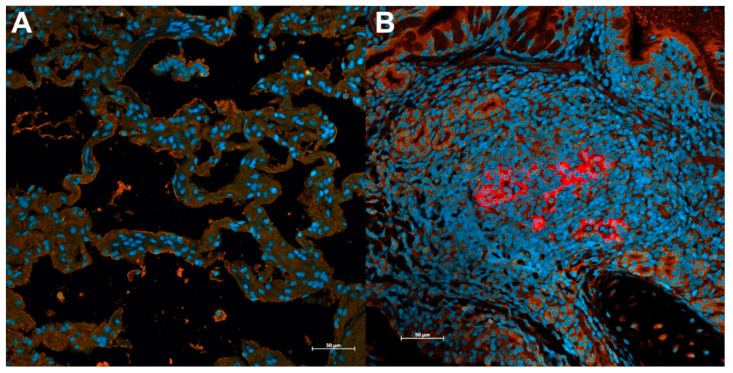
Immunofluorescence Assay (IFA) of SARS-CoV-2 lung infection in domestic cats. (**A**) Normal lung showing DAPI-stained nuclei (blue) and normal ACE2-expression (orange) on the surface of lung epithelial cells. (**B**) Anti-SARS-CoV-2 nucleocapsid protein (red) is present within peribronchial glandular epithelial cells expressing the ACE2 receptor (orange) in CoV-2 infected animals at 4 dpi. Blue = DAPI/nuclei; Red = SARS-CoV-2 antigen; Orange = ACE2 receptor. Magnification 20×, scale bar = 50 µm.

**Figure 5 viruses-14-01207-f005:**
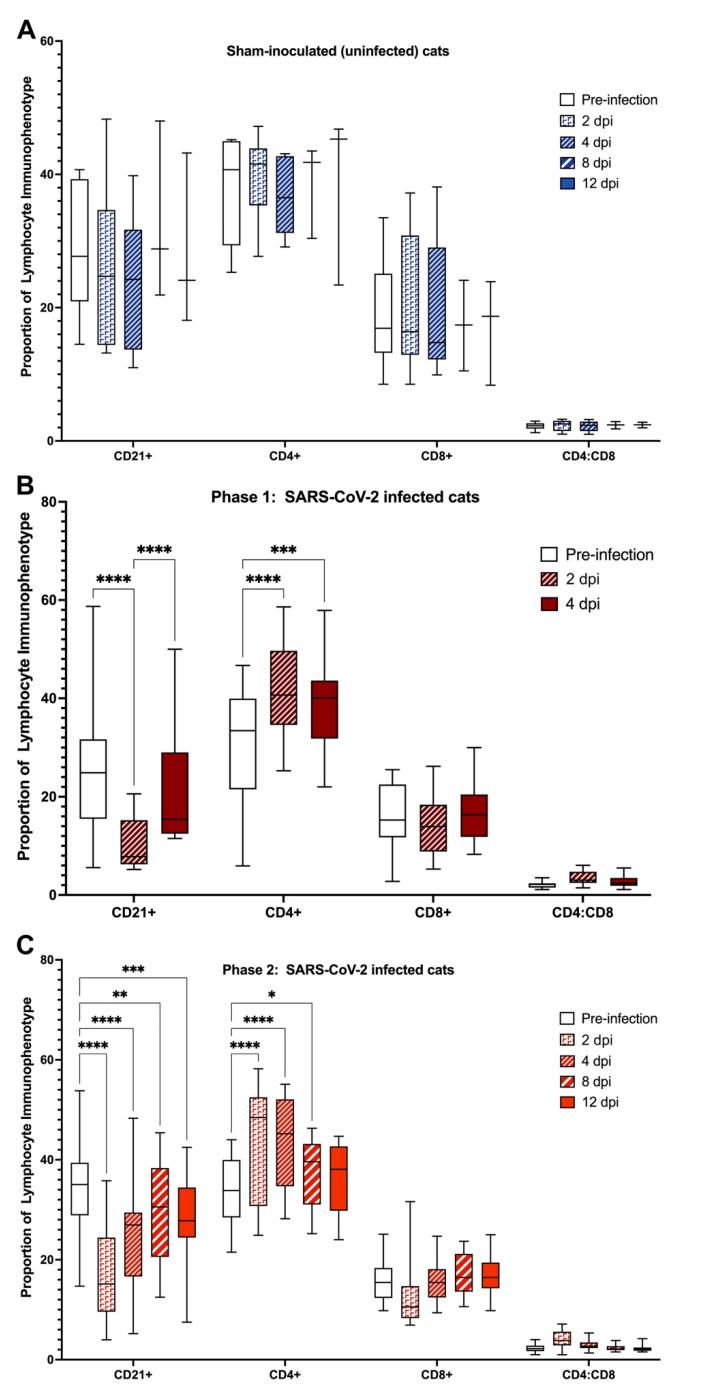
SARS-CoV-2 infected cats exhibit divergent systemic immunophenotypes. (**A**) Sham-inoculated controls do not exhibit any significant differences in lymphocyte immunophenotype throughout the course of the study. (**B**) In contrast, SARS-CoV-2 infected cats in Phase 1 exhibit a significant decrease in the proportion of CD21+ cells at both 2 and 4 dpi. This shift is accompanied by a simultaneous increase in the proportion of CD4+ cells at 2 and 4 dpi. (**C**) Similar to Phase 1, SARS-CoV-2 infected cats exhibit different shifts in systemic lymphocyte immunophenotypes, characterized by a significant decrease in CD21+ cells at 2, 4, 8, and 12 dpi with a concurrent increase in CD4+ cells at 2, 4, and 8 dpi. Repeated measures ANOVA with multiple comparisons (Tukey); * *p* < 0.05, ** *p* < 0.01, *** *p* < 0.001, **** *p* < 0.0001.

**Figure 6 viruses-14-01207-f006:**
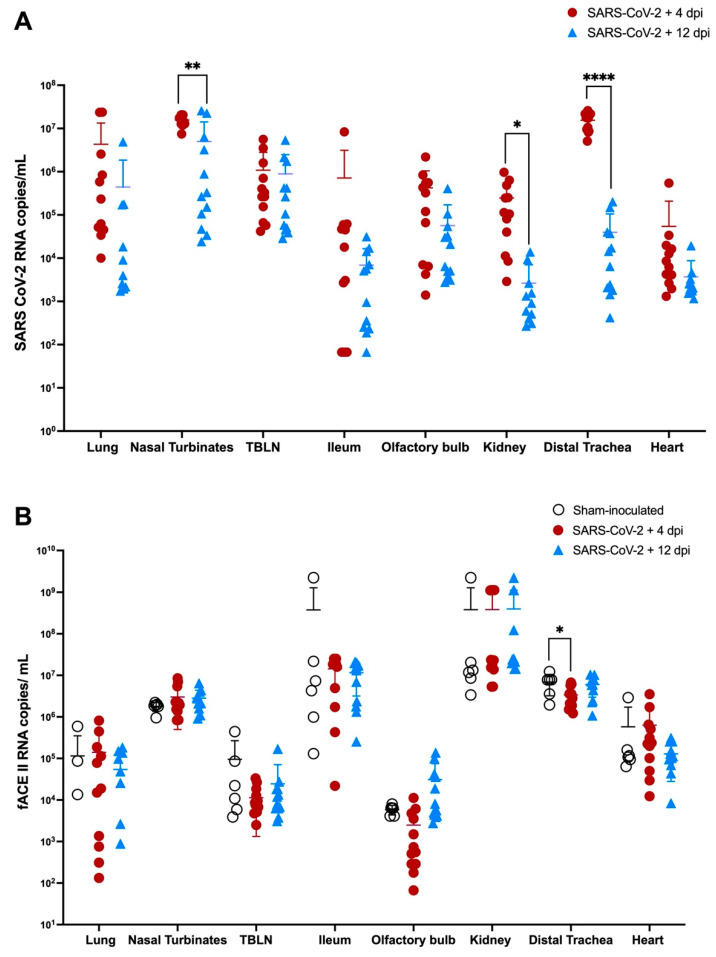
SARS CoV-2 viral RNA and f-ACE2 RNA quantification in infected feline tissues. (**A**) SARS CoV-2 viral copies are quantified from all the tissues collected at 4 and 12 dpi from SARS CoV-2 infected cats. Viral loads in the distal trachea, kidney, and nasal turbinates were significantly higher at 4 dpi compared to 12 dpi during the SARS-CoV-2 Delta variant. Viral RNA copies were also slightly increased in the olfactory bulb at 4 dpi compared to 12 dpi (*p* = 0.08) (**B**) f ACE2 receptor RNA is significantly decreased in the distal trachea compared to sham-inoculated controls (*p* = 0.0456). Data are expressed as means ± SD (*n* = 12) cats per group. Tissues from sham-inoculated cats had no detectable virus. Statistical comparisons are made via one-way ANOVA. * *p* < 0.05, ** *p* < 0.01, **** *p* < 0.0001.

**Figure 7 viruses-14-01207-f007:**
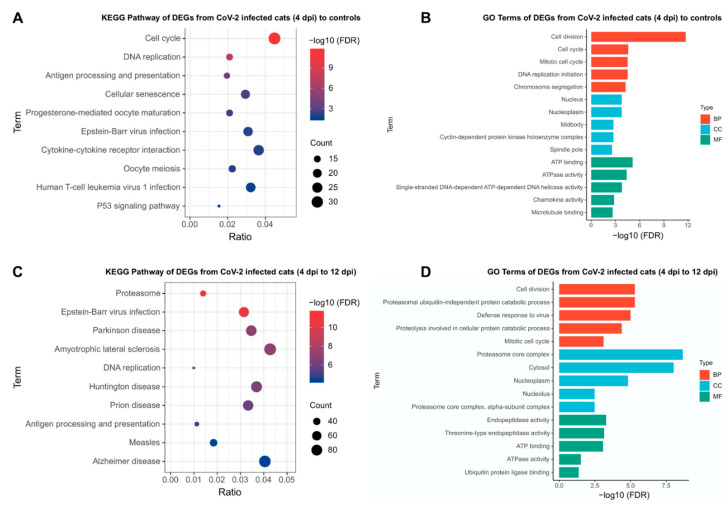
KEGG pathway and gene ontology (GO) terms enrichment in lung samples during SARS-CoV-2 infection. KEGG pathway and gene ontology (GO) terms enrichment based on DEGs in uninfected controls vs. 4 dpi (**A**,**B**) and 4 dpi vs. 12 dpi (**B**,**D**). KEGG Pathway analysis (**A**,**C**) with DEG’s denotes the ratio of the number of differentially expressed genes (x-axis) compared to the total genes in a particular pathway (y-axis). Gene ontology (GO) enrichment analysis (**B**,**D**) denotes the algorithm of adjusted p-values (FDR, x-axis) corresponding to specific GO pathways (y-axis).

**Figure 8 viruses-14-01207-f008:**
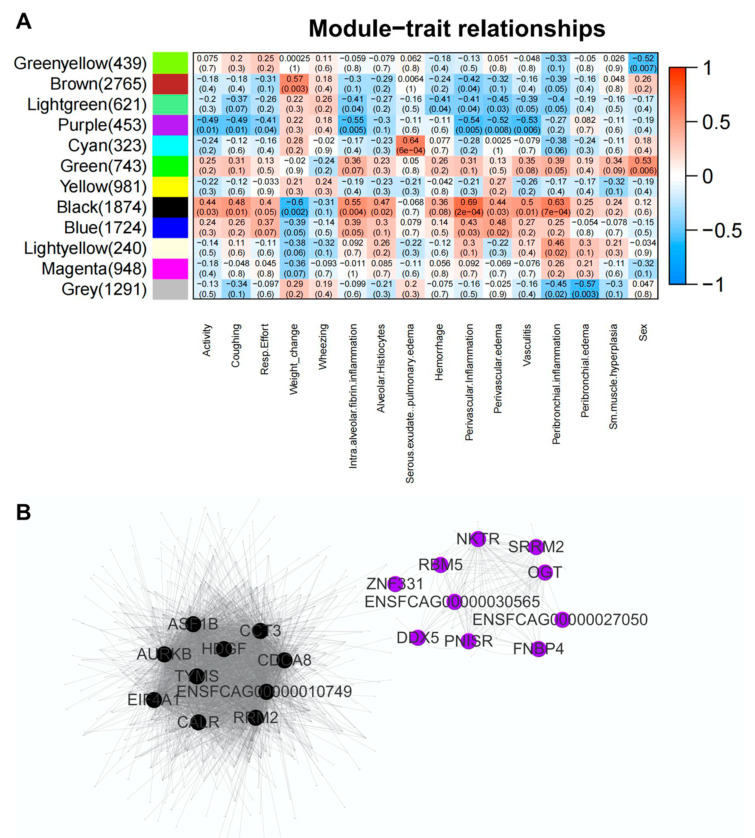
Weighted correlation network analysis (WGCNA) on lung RNA-Seq data obtained from SARS-CoV-2 infected domestic cats. (**A**) Relationships of consensus module eigengenes and histopathology and clinical traits. Each row in the figure corresponds to a consensus module, and each column to a histopathological trait. Numbers in the cells report the correlations of the corresponding module eigengenes and traits, with the *p*-values printed below the correlations in parentheses. The cell is color coded by correlation according to the color legend. (**B**) Identification of hub genes based on intra-modular connectivity in black and purple modules.

**Table 1 viruses-14-01207-t001:** Clinical Scoring System for Feline Respiratory Disease, adapted from [26].

Clinical Parameter	0 (Healthy)	1	2	3
Body Weight	No weight loss	0 to 5% weight loss	5 to 10% weight loss	>10% weight loss
Temperature	37.2 to 39.2 °C	39.1 to 39.4 °C	39.5 to 39.7 °C	>39.7 °C
SpO_2_	98 to 100%	95 to 97%	93 to 94%	<93%
Activity	Normal	Mild reduction when disturbed * (mild lethargy)	Moderate reduction when disturbed * (moderate lethargy)	Little to no activity disturbed * and reduced activity when stimulated **
Behavior	Normal	Reduced interest in food and/or attention	Markedly reduced interest in food and/or attention	Anorexia and/or complete lack of interest
Respiratory Effort	Normal resting respiratory rate and normal effort	Mild tachypnea (>35 breaths per minute); no overt increase in effort	Moderate tachypnea (>40 breaths per minute); moderate increase in effort	Marked tachypnea (>45 breaths per minute); marked effort or dyspnea
Ocular and/or Nasal Discharge	None	Mild discharge noted	Moderate discharge noted	Marked discharge noted
Coughing	None	Occasional, rare cough	Intermittent coughing (at least one episode in 30 min)	Marked persistent coughing(2 or more episodes over 30 min)
Wheezing	None	Occasional, rare wheezing	Intermittent wheezing	Marked, frequent wheezing

* Disturbed: observer in the room but kennel unopened. ** Stimulated: kennel open.

**Table 2 viruses-14-01207-t002:** Sequencing and Assembly Data.

ISOLATE ID	Inoculation Days	Total Number of Reads	Genome Coverage	% Draft Genome Assembled
Viral RNA	Day 0	724,000	7264.99×	97.71%
9671 Cr. Lung	Day 4	660,000	6622.52×	95.5%
9671 Tbln	Day 4	524,000	5257.88×	96.55%

**Table 3 viruses-14-01207-t003:** A synonymous mutation was detected in ORF 1ab at nucleotide position 13131.

Nt Position	Genomic Region	hCoV-19/USA/PHC658/2021	Viral RNA	Lung	TBLN	Mutation Type
13131	ORF 1ab	A	A	G	A	Synonymous

## Data Availability

Raw sequencing data reads are deposited in FASTQ format to the NCBI Sequence Read Archive database (SRA) under a BioProject accession number PRJNA842733.

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
