# Peer review of "SARS CoV-2 (Delta Variant) Infection Kinetics and Immunopathogenesis in Domestic Cats"

_viruses, 2022, doi:10.3390/v14061207_

Round 1

Reviewer 1 Report

In this study the authors investigate in detail the infection and disease dynamics of the SARS-CoV-2 Delta variant in experimentally infected domestic cats. Two different routes of inoculation (IT, and IT+IN), and immunopathology at 2 time points (4 and 12 DPI) are compared. In addition to clinical disease, virus shedding and pathological lesions, the authors look at blood immunophenotypes and transcriptomic analysis in the lungs.  The study is thorough and presented well.  While these studies suggest intratracheal inoculation of domestic cats with SARS-CoV-2 may serve as a potential model for COVID research, there remains ethical concerns using companion animals for research. 

Minor comments:

Line 186 says "serial culture of nasal swabs collected at all timepoints". Does "serial culture" mean these were serial passaged, and if so how many times? Line 175 says 6 dpi samples were collected; were 6 dpi tested and if so what were those results?  Also it is not clear how many nasal swab samples were VI positive.  In the Fig. S4 graph the axis says % positive CPE; is this % positive animals out of those tested or?  Please clarify number of virus positive swabs identified on each of the days tested. 

What was the percentage of this mutation in ORF 1ab 13131? This was only detected in one animal lung?

Line 390: add a “.”

Line 557-560:  is this sentence referring to Sham group? Please clarify.

Line 625:  should this read, “…which was NOT statistically significant.”?  Or does the Fig. 6B need modified for heart and lung?

Line 739: spelling “after”

In several spots “supplemental data” is indicated; it would be helpful if the supplemental figures were specified.

There is a report of a symptomatic cat naturally infected with Delta from Spain, which might be of interest to include in the discussion:   Front. Vet. Sci., 01 April 2022, https://doi.org/10.3389/fvets.2022.841430

Author Response

Response to Reviewers

Reviewer #1

In this study the authors investigate in detail the infection and disease dynamics of the SARS-CoV-2 Delta variant in experimentally infected domestic cats. Two different routes of inoculation (IT, and IT+IN), and immunopathology at 2 time points (4 and 12 DPI) are compared. In addition to clinical disease, virus shedding and pathological lesions, the authors look at blood immunophenotypes and transcriptomic analysis in the lungs.  The study is thorough and presented well.  While these studies suggest intratracheal inoculation of domestic cats with SARS-CoV-2 may serve as a potential model for COVID research, there remains ethical concerns using companion animals for research. 

Thank you so much for the insightful summary and encouraging comments for the manuscript.

Minor comments:

Line 186 says "serial culture of nasal swabs collected at all timepoints". Does "serial culture" mean these were serial passaged, and if so how many times? Line 175 says 6 dpi samples were collected; were 6 dpi tested and if so what were those results?  Also it is not clear how many nasal swab samples were VI positive.  In the Fig. S4 graph the axis says % positive CPE; is this % positive animals out of those tested or?  Please clarify number of virus positive swabs identified on each of the days tested. 

We appreciate your comments. We collected nasal swab samples at each time point and cultured the samples for infectious virus.  To convey this content more accurately, the word ‘serial’ has been replaced with ‘cell’. The error of including a ‘6 dpi’ was corrected in the context. Blood and nasal swab samples were not collected at this timepoint, and this time point has therefore been removed from the text. All the infected cats (24/24 cats) had positive nasal swab results until 4 dpi. This has been further clarified in the results section of - Infectious virus is shed via nasal secretions up to 4-days post inoculation (Line 417)

What was the percentage of this mutation in ORF 1ab 13131? This was only detected in one animal lung?

Thanks for the suggestion. Due to resource constraints, sequencing was only performed in lung and lymph node tissues of a single animal and the mutation was only detected in the lung tissue. Since the data is available only for a single animal, it is not possible to depict it as a percentage.

Line 390: add a “.”

The sentence was amended accordingly

Line 557-560:  is this sentence referring to Sham group? Please clarify.

Thank you for this oversight. This sentence is referring to SARS-CoV-2 infected cats and was edited in the manuscript for clarification.

Line 625:  should this read, “…which was NOT statistically significant.”?  Or does the Fig. 6B need modified for heart and lung?

Thank you for highlighting this error. The text has been corrected accordingly.

Line 739: spelling “after”

The spelling error was corrected in the manuscript.

In several spots “supplemental data” is indicated; it would be helpful if the supplemental figures were specified.

Thank you for the suggestion. The “supplemental data” indicated in the texts refers to raw data and statistical output data that are included as supplemental data files and does not correlate with particular supplemental figures, unless specifically noted.

There is a report of a symptomatic cat naturally infected with Delta from Spain, which might be of interest to include in the discussion:   Front. Vet. Sci., 01 April 2022, https://doi.org/10.3389/fvets.2022.841430

Thank you for the excellent suggestion. This reference has been added in the discussion section along with a brief synopsis.

Reviewer 2 Report

Manuscript ID: viruses-1737675

SARS CoV-2 (Delta variant) infection kinetics and immuno-2 pathogenesis in domestic cats

Manuscript Type: Article

General Comments: The present manuscript presents a detailed study of the pathogenesis of the SARS-CoV-2 Delta variant in domestic cats. The study is well-designed and the results are clearly presented. The thoroughness of the study provides an in depth characterization of the pathogenesis at gross, tissue, cellular and transcriptional levels. As such, the authors make a compelling case for the use of the feline model to further investigate other SARS-CoV-2 variants as well as to explore interventions. No major concerns were identified but several comments follow to assist in improving the manuscript.

Specific Comments:

Line 117: Please indicate whether the cats were intact or neutered/spayed.

Line 177: Were complete blood counts performed? If so please state whether any abnormalities were identified.

Line 551: Please report the absolute cell numbers rather than percentages. The percentages do no indicate whether there is an actual increase or decrease of any particular lymphocyte subset and thus cannot be interpreted as stated. The corresponding section in the discussion will also need revision.

Line 627 and throughout: The term dysregulated is used throughout the manuscript in reference to differences in RNA expression between groups. It would be more appropriate to simply indicate increased/decreased as actual ‘dysregulation’ of expression is not shown. It is more likely that gene transcription is appropriate in response to the pathogen and tissue damage. It is also likely that different cell populations are being assessed in the normal tissues of the control animals versus the inflamed tissues of the infected animals revealing activation of transcription unique to the inflammatory cell population. Were type I interferons increased in infected tissues?

Line 739: Correct the spelling of after.

Line 744: This conclusion/association cannot be made from the data that is presented. No evidence of T cell inflammation or a Th1 response is shown.

Line 754: There are a number of spacing issues throughout manuscript. A final proof read by the authors will be needed to correct these.

Line 762: There seems to be a word missing in this sentence.

Line 800: It is not clear how AURKB might be related to N protein mutations. Please provide additional information.

Line 812: Please clarify this sentence.

Figure 7: The legend is insufficient to interpret the data. Please define the x-axis abbreviation and explain the legend for A and C.

References: The format is highly variable and sometimes incomplete. The references  need to be carefully reviewed by the authors. Bosco-Lauth et al. is listed three times.

Author Response

Response to Reviewers

Reviewer #2

The present manuscript presents a detailed study of the pathogenesis of the SARS-CoV-2 Delta variant in domestic cats. The study is well-designed and the results are clearly presented. The thoroughness of the study provides an in depth characterization of the pathogenesis at gross, tissue, cellular and transcriptional levels. As such, the authors make a compelling case for the use of the feline model to further investigate other SARS-CoV-2 variants as well as to explore interventions. No major concerns were identified but several comments follow to assist in improving the manuscript.

We appreciate your effort in thorough evaluations and positive suggestions for the manuscript.

Specific Comments:

Line 117: Please indicate whether the cats were intact or neutered/spayed.

Thank you for the suggestion. The cats were neutered / spayed, and this has been included in the text.

Line 177: Were complete blood counts performed? If so please state whether any abnormalities were identified.

Thank you for this comment. Since all blood samples were collected in a BSL-3 facility, samples were required to be neutralized before leaving BSL-3 facility. Conducting complete blood count (CBC) assays or serum chemistry analysis through a reference laboratory was not possible due to the infectivity of samples and inability to perform these tests following virus neutralization protocols.  Moreover, instrumentation to perform CBCs was not available in the BSL-3/ABSL-3 laboratories, thus inclusion of CBS assays was conducted for this study..

Line 551: Please report the absolute cell numbers rather than percentages. The percentages do no indicate whether there is an actual increase or decrease of any particular lymphocyte subset and thus cannot be interpreted as stated. The corresponding section in the discussion will also need revision.

Due to the constraints of working within our BSL-3 facility (see above) and lack of available instrumentation, CBC analysis was not able to be performed and total lymphocyte counts were not obtained for blood samples from infected cats. This prevented us from back calculating the total number of lymphocytes for each specific lineage in each sample, especially since the cell counts in each sample varied by individual on flow cytometry. Alterations in the proportion of lymphocyte populations have been previously reported in cats by our lab [1] and by several other labs [2-5] as a way to compare shifts in circulating lymphocyte immunophenotype during feline viral infections. Based upon this standardized practice and under the constraints of working with BSL-3 infectious samples, it is the best we can do.

  1. Miller, C.; Powers, J.; Musselman, E.; Mackie, R.; Elder, J.; VandeWoude, S. Immunopathologic Effects of Prednisolone and Cyclosporine A on Feline Immunodeficiency Virus Replication and Persistence. Viruses 2019, 11, doi:10.3390/v11090805.
  2. Mexas, A.M.; Fogle, J.E.; Tompkins, W.A.; Tompkins, M.B. CD4+ CD25+ regulatory T cells are infected and activated during acute FIV infection. Veterinary immunology and immunopathology 2008, 126, 263-272.
  3. Joshi, A.; Vahlenkamp, T.W.; Garg, H.; Tompkins, W.A.; Tompkins, M.B. Preferential replication of FIV in activated CD4+ CD25+ T cells independent of cellular proliferation. Virology 2004, 321, 307-322.
  4. Miranda, L.H.M.d.; Meli, M.; Conceição-Silva, F.; Novacco, M.; Menezes, R.C.; Pereira, S.A.; Sugiarto, S.; Reis, É.G.d.; Gremião, I.D.F.; Hofmann-Lehmann, R. Co-infection with feline retrovirus is related to changes in immunological parameters of cats with sporotrichosis. PloS one 2018, 13, e0207644.
  5. de Groot-Mijnes, J.D.; Van Der Most, R.G.; Van Dun, J.M.; Te Lintelo, E.G.; Schuurman, N.M.; Egberink, H.F.; De Groot, R.J. Three-color flow cytometry detection of virus-specific CD4+ and CD8+ T cells in the cat. Journal of immunological methods 2004, 285, 41-54.

Line 627 and throughout: The term dysregulated is used throughout the manuscript in reference to differences in RNA expression between groups. It would be more appropriate to simply indicate increased/decreased as actual ‘dysregulation’ of expression is not shown. It is more likely that gene transcription is appropriate in response to the pathogen and tissue damage. It is also likely that different cell populations are being assessed in the normal tissues of the control animals versus the inflamed tissues of the infected animals revealing activation of transcription unique to the inflammatory cell population. Were type I interferons increased in infected tissues?

Thank you for the suggestion. As a result of our RNA Seq analysis, we identified over 1,365 genes that were upregulated during SARS-CoV-2 infection, and 1,607 genes that were downregulated.  We have listed all of these genes in our supplemental data but to not have the space to address which genes are upregulated versus downregulated.  The term “dysregulated” is preferred in this instance to serve as an encompassing term to denote shifts in gene expression that may prove useful in future studies of SARS-CoV-3 pathogenesis. To keep the discussion concise, we have selected a few examples of genes and described their alterations in regulation (upregulated versus downregulated).  Future studies may utilize our robust data set to investigate potential therapeutic targets for downstream analysis, but that is beyond the scope of this project. Additionally, we completely agree that different cell populations may have been assessed in the normal tissues of the control animals versus the inflamed tissues of the infected animals (revealing activation of transcription unique to the inflammatory cell population), and future studies will seek to utilize single cell RNA Seq to compare transcription profiles within specific cell types.  However, that technology was not available to us nor within the constraints of the BSL-3 environment, and so bulk RNA Seq was performed.  Although not optimal, this assessment provides a robust data set to guide future studies in transcriptomics of SARS-CoV-2 infection in cats and how they compare with human COVID-19.

Line 739: Correct the spelling of after.

Thanks for identifying this. The spelling error has been corrected.

Line 744: This conclusion/association cannot be made from the data that is presented. No evidence of T cell inflammation or a Th1 response is shown.

Thank you for the comment. This is not a conclusion per say, and is theoretical speculation based on previous studies (see references). This has been updated in the discussion section to more accurately convey our intent.

Line 754: There are a number of spacing issues throughout manuscript. A final proof read by the authors will be needed to correct these.

Thank you for this comment. The spacing issues have been corrected as per suggestions.

Line 762: There seems to be a word missing in this sentence.

Thank you for bringing this to our attention. This text has been amended.

Line 800: It is not clear how AURKB might be related to N protein mutations. Please provide additional information.

Thank you for this comment. Additional information has been included as suggested.

Line 812: Please clarify this sentence.

Thank you for the suggestion. This information has been added to the text

Figure 7: The legend is insufficient to interpret the data. Please define the x-axis abbreviation and explain the legend for A and C.

Thanks for the suggestion. Further explanation along with the defined x-axis abbreviation for the Fig 7 has been added in the manuscript.

References: The format is highly variable and sometimes incomplete. The references need to be carefully reviewed by the authors. Bosco-Lauth et al. is listed three times.

Thank you for catching this! The references were carefully reviewed and cross-checked by the authors as suggested and duplicates were removed.
